# Evaluation of Hydroclimatic Variability and Prospective Irrigation Strategies in the U.S. Corn Belt [†]

**María Elena Orduña Alegría [1,*], Niels Schütze [1] and Dev Niyogi [2]**

[1]   Institute of Hydrology and Meteorology, Technische Universität Dresden, Bergstr. 66, 01069 Dresden, Germany; niels.schuetze@tu-dresden.de

[2]   Department of Agronomy and Department of Earth, Atmospheric, and Planetary Sciences, Purdue University, 550 Lafayette St, West Lafayette, IN 47907, USA; climate@purdue.edu

*   Correspondence: maria_elena.orduna_alegria@tu-dresden.de

†   This paper is an extended version of our paper published in the EGU General Assembly 2019, Vienna, Austria, 7–12 April 2019.

**Abstract:** Changes in climate, land use, and population growth has put immense pressure on the use of water resources in agriculture. Non-irrigated fields suffer from variable water stress, leading to an increase in the implementation of irrigation technologies, thus stressing the need to analyze diverse irrigation practices. An evaluation of 17 sites in the U.S. Corn Belt for two temporal climatic conditions was carried out. It consisted of the analysis of critical hydroclimatic parameters, and the evaluation of seven diverse irrigation strategies using the Deficit Irrigation Toolbox. The strategies included rainfed, full irrigation, and several optimizations of deficit irrigation. The results show a significant change in the hydroclimatic parameters mainly by increased temperature and potential evapotranspiration, and a decrease in precipitation with an increase in intense short rainfall events. Consequently, the simulations indicated the potential of deficit irrigation optimization strategies to increase water productivity above full irrigation and rainfed conditions. In particular, GET-OPTIS for wet soil conditions and the Decision Tables for dry soil conditions seasons. The present study highlights the contributions of atypical weather to crop production and the implications for future management options, and allows specialized regionalization studies with the optimal irrigation strategy.

**Keywords:** crop-water productivity; irrigation strategy optimization; agroclimatic resiliency; crop-climate decision tools; U.S. Corn Belt

## 1. Introduction

The spatial and temporal variability of climate, land use, soil degradation, and population growth put immense pressure on water resources. Sustainability and resilience depend strongly on the way managers and consumers adapt to the current and predicted variability. In particular, the intense pressure on food security hinders adequate water resource management, primarily in the face of rainfall vagaries and when agriculture relies on or is expected to rely on irrigation. The agrohydrological dilemma (i.e., securing food production in water scarcity scenarios) was analyzed in several studies focusing on the impacts of climate variability on crop yield (e.g., Niyogi et al. [1], Brumbelow et al. [2], Rosenzweig et al. [3] and Elliot et al. [4]). Studies such as Pereira [5] and Gorantiwar et al. [6] focused on the improvement of irrigation techniques, while Dobernmann et al. [7], Godfray et al. [8] and Rockstrom et al. [9] focused on the prospective future of sustainable agriculture through irrigation availability.



This study focuses on the agricultural production of corn (*Zea mays L.*) in the Corn Belt region of the United States of America (US). Crop yields in the Corn Belt were projected to go down in the future climate as a result of an increase in extreme weather events and increased rainfall variability [10].

Irrigation has the potential to become a globally implemented adaptation strategy in the face of climate change. In the simplest sense, irrigation practices seek to apply water to the soil and plant for effective crop production by influencing stages from germination to yield. Crop simulation models allow investigating outcomes for different management schemes that might increase the yield [11]. One of the main tools to achieve this is irrigation scheduling (a sequence of dates or and times on which water needs to be applied to the crop), can be optimized by mathematical models [12]. A common irrigation strategy, known as full irrigation, is to supply sufficient water to meet with the plant evapotranspiration requirements. Due to the scarce nature of water resources, other strategies were developed, such as supplemental and deficit irrigation, to reduce the agricultural water demand and to divert the resources for alternative uses. Supplemental irrigation is the application of small amounts of water to rainfed crops when rainfall does not meet the plant evapotranspiration requirements, and deficit irrigation is the optimized application of water below the plant evapotranspiration requirements. Both irrigation strategies were thoroughly analyzed and optimized to maximize water productivity and to maintain yields [13,14]. Crop Water Productivity (CWP), defined as crop yield per cubic meter of water consumption [15] is a good indicator of water-agriculture interaction. The CWP function can be used to show the obtainable yield at different levels of applied water. The CWP functions (CWPFs) are characterized by linearly increasing yields with applied water until 50% of full irrigation [16–18]. The relationship becomes curvilinear as applied water increases further, due to losses from increased surface evaporation, runoff and deep percolation. Moreover, local factors, such as soil and irrigation technology, can affect the relation [19]. Furthermore, climate variability has an impact on CWPFs, which highlights the importance of a stochastic approach to irrigation [2]. Recent studies by Evett et al. [11], Raju et al. [12], English et al. [19], Brown et al. [20] and Shang et al. [21] indicate that a detailed and precise irrigation schedule calculated using crop models can optimize the CWP by maximizing irrigation efficiency, reducing costs and environmental impacts. Irrigation scheduling [22] is conventionally based on soil water balance models, where the soil moisture deficit is estimated by the difference between the inputs (irrigation and precipitation) and the losses (runoff, percolation, and evapotranspiration). The adequate water volume to be irrigated varies as a function of actual evaporative demand, for deficit irrigation strategies this is a complex task to achieve because of the day to day variation in climate and crop water demands. The impact of hydroclimatic variability was investigated (e.g., Djaman et al. [23], Badh et al. [24], Gunn et al. [25], Messina et al. [26], Niyogi et al. [27], Panagopoulos et al. [28] and Zwart et al. [29]), and deficit and supplemental irrigation strategies are often promoted as a response to mitigate drought stress on crops [6,13,14,30–32]. However, very few studies evaluated different irrigation strategies in the same location as a measure of hydroclimatic variability and sustainable agricultural productivity. Studies by Niyogi et al. [1], Yang et al. [33], Song et al. [34], and Kloss et al. [31] highlight the ability of crop models to capture the impacts of climate variability on yield considering different sources of uncertainty. Most of the crop models aim to achieve an optimum water supply for productivity, with soil water content being maintained close to field capacity, most commonly via conventional or supplemental irrigation (i.e., 100% of field capacity) [35]. Alternatively, deficit irrigation strategies were developed as an adaptation to limited water availability by estimating the supply of irrigation during the most sensitive growth stages and allowing prioritization of the allocation of resources to these drought-sensitive stages [10,32]. Deficit irrigation strategies aim for a determined lower percentage, typically between 70%–90%, of field capacity [13,30]. The optimal time to irrigate depends on the seasonal water demand pattern which varies by crop, the hydraulic soil characteristics, and the available amount of water [36]. The estimation of the irrigation scheduling is aimed to obtain the highest potential crop yield for a given total seasonal depth of irrigation. However, these estimations are also limited by preconditions of access to a perfect forecast of intraseasonal crop water requirements [37]. As an alternative to such

idealized consideration, optimization approaches based on decision tables or a framework such as the Optimal Climate Change Adaption Strategies on Irrigation Methodology (OCCASION) [38] are available.

Most of the simulation-based studies of deficit irrigation do not consider the variability of important climate parameters, i.e., temperature, evapotranspiration and precipitation, within different temporal scales. The studies mostly focused on all rainfed sites or at irrigated sites with assumption about full field capacity irrigation [14,31,32,39,40]. This highlights the need for multidisciplinary simulations where different irrigation management strategies for corn production are compared and assessed. Therefore, based on the projected changes in water resources availability and the potential of implementation of irrigation technologies in the intense agriculture in the Corn Belt, the objective of this study was to understand the hydroclimatic variability at different temporal scales and to evaluate supplemental and deficit irrigation optimizers under potential water scarcity conditions over locations across the US Corn Belt.

## 2. Materials and Methods

### 2.1. Study Area

Corn, the primary US feed grain, accounts for around 500–600 billion tons of production in the US [41,42]. Most of the corn production occurs in the Corn Belt, a region in the US Midwest known for the ideal climate and soil conditions for crop production and intense farming characterized by high fertile soils, high organic soil concentration, timely rainfall, and ample solar radiation. Geographically, the Corn Belt consists of the states of Iowa, Illinois, Indiana, Nebraska, Kansas, Minnesota, Missouri, South Dakota, North Dakota, Ohio, Wisconsin, and parts of Michigan and Kentucky. The region is divided by two large intensively cropped river basins, the Upper Mississippi River Basin and Ohio-Tennessee River Basin and it is located within five water resources regions (Missouri, Arkansas-White-Red, Souris-Red-Rainy, Upper Mississippi, Lower Mississippi, Ohio, and the Great Lakes) [28].

County-level data of corn yield and climatic variables were assessed and used following Niyogi et al. [1] and Liu et al. [43]. This provided a spatially representative data set for 17 sites within the US Corn Belt. Information regarding these sites is provided in Figure 1 and Table 1.

**Table 1.** Summary of the 17 study sites.

| # | Code | Site | County | State | Area Harvested [$\times$1000 ha] | Irrigated Area [%] |
|---|------|------|--------|-------|----------------------------------|--------------------|
| 1 | W1 | Kirksville | Adair | Missouri | 5.73 | NDD |
| 2 | W2 | Topeka | Shawnee | Kansas | 15.29 | 31 |
| 3 | W3 | New Madrid | New Madrid | Missouri | 27.51 | 79 |
| 4 | W4 | Olivia | Renville | Minnesota | 43.97 | <0.1 |
| 5 | W5 | Brookings | Brookings | South Dakota | 47.87 | 8 |
| 6 | W6 | Iowa City | Johnson | Iowa | 55.44 | NDD |
| 7 | W7 | Grand Forks | Grand Forks | North Dakota | 56.30 | 4 |
| 8 | W8 | Columbus | Platte | Nebraska | 75.72 | 67 |
| 9 | W9 | Rochester | Olmsted | Minnesota | 115.32 | <0.1 |
| | | | | Total | 443.16 | 19 |
| 10 | E1 | Marysville | Union | Ohio | 8.88 | NDD |
| 11 | E2 | Toledo | Lucas | Ohio | 29.02 | NDD |
| 12 | E3 | Huntington | Huntington | Indiana | 30.41 | <1 |
| 13 | E4 | Baraboo | Sauk | Wisconsin | 32.65 | 19 |
| 14 | E5 | DeKalb | DeKalb | Illinois | 50.44 | <0.01 |
| 15 | E6 | Beloit | Rock | Wisconsin | 60.59 | 7 |
| 16 | E7 | Rensselaer | Jasper | Indiana | 62.99 | 9 |
| 17 | E8 | Tuscola | Douglas | Illinois | 104.2 | <1 |
| | | | | Total | 379.19 | 5 |

NDD: not disclosed data; 1 ha = 10,000 m$^2$.

Figure 1 shows a representative county outline map of the US Corn Belt with the distribution of irrigation intensity. The counties with the most irrigated area are in the southwest of the Corn Belt, and the center or eastern region is almost entirely rainfed agriculture with no irrigation reported. The study sites (Table 1) are divided into two parts across the Corn Belt, the Western (red) and Eastern (blue). The sites in the Western Corn Belt reported more use of irrigation technologies with two counties (New Madrid, MO, and Platte, NE) with more than 60% irrigated area. On the contrary, the sites located in the Eastern Corn Belt have mainly rainfed agriculture with less than 10% irrigated area with the exception of Baraboo, WI. This irrigation intensity can be considered representative of the ground reality across the Corn Belt.

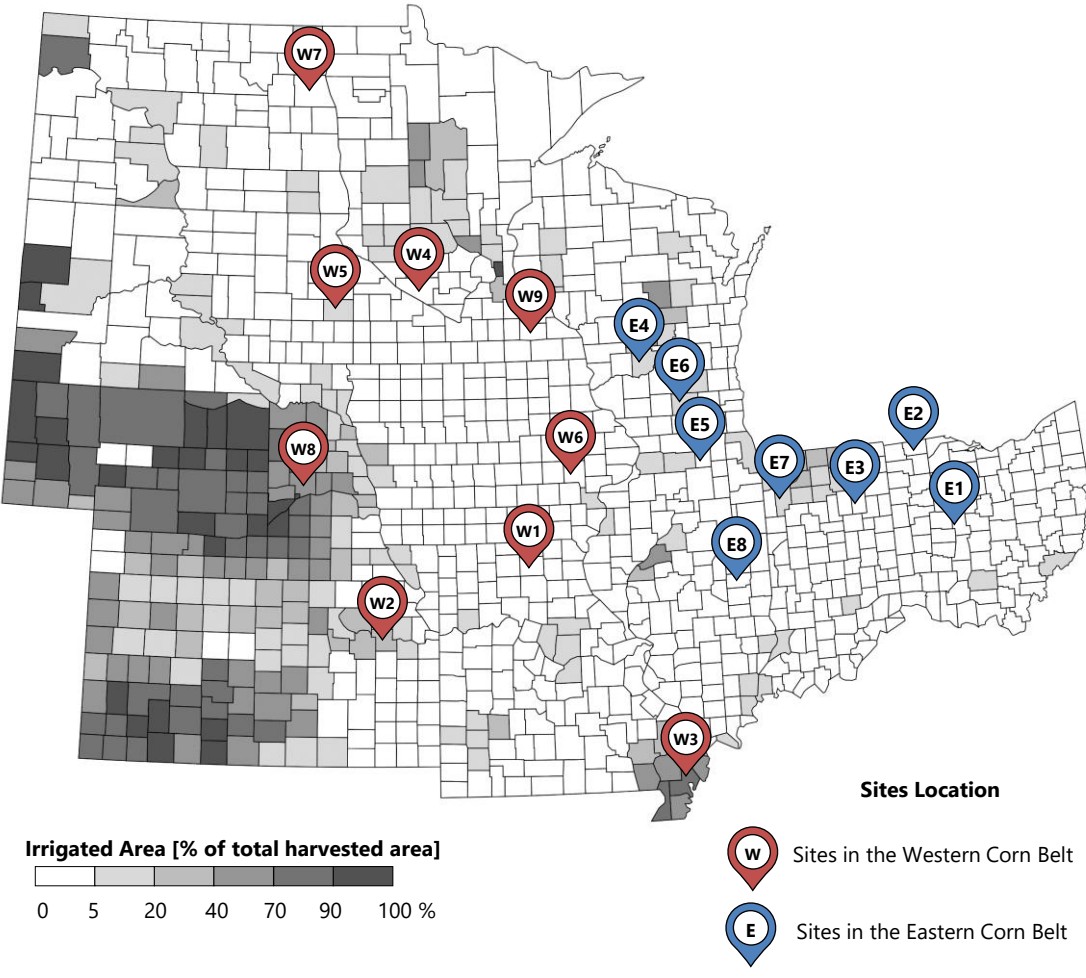

**Figure 1.** Map showing the reported irrigated area by county along in the US Corn Belt and the location of 17 study sites (2012 NASS-USDA [44]).

## 2.2. Data

To analyze the impacts of climate variability on crop yield, historical (1981–2010) and future climatic conditions (2041–2070) were considered for the 17 study sites. The data required was compiled as input the modeling framework discussed in Section 2.4, details can be found in Supplementary Material SI. The data sets included were the daily meteorological data (minimum temperature, maximum temperature, and precipitation) for the historical climatic conditions from the National Centers for Environmental Information (NCEI, [1]). The future climate condition was obtained from the National American Regional Climate Change Assessment Program (NARCCAP, [45]) from the

dynamically downscale product from the MM5 with the Hadley Centre Climate Model version 3. Further information about this data set can be found in Mearns et al. [46–48], and Horton et al. [49]. The irrigation strategy model requires information about evapotranspiration water loss. Because of the lack of this information for the historical climatic conditions and to keep consistency between different time scales, the daily potential evapotranspiration was calculated with the FAO ETo Calculator [50] using the Penman-Monteith equation. Additional agronomic information was required which was compiled from the National Corn Handbook [51], which included the extent of the growing season of around 130 to 150 days across the Corn Belt. A period of 150 days was considered and divided into four phenological stages. These stages included: initial planting/germination (30 days), crop development (40 days), mid-season (50 days) and late development (30 days). The growing season dates were specifically chosen for each site from the reported dates in the Field Crops Usual Planting and Harvesting Dates [52], for the historical climatic conditions from 1997 and for the future climatic conditions from 2010.

*2.3. Irrigation Strategies*

When this study was conducted, only seven irrigation strategies were integrated into the Deficit Irrigation Toolbox (DIT) described in Section 2.4. The present study seeks to analyze rainfed, full and deficit irrigation strategies with diverse management optimization to assess these strategies relative to the on-going practices in the study sites. Therefore all the seven irrigation strategies were considered. These include (i) no irrigation (rainfed system), (ii) full (supplemental) irrigation, and (iii) five deficit irrigation strategies. These seven strategies were:

1. **Rainfed** (**S1_RF**): consists of no water application to simulate rainfed agriculture. This is used as a reference and is expected to produce a lower limit of yields.
2. **Full supplemental irrigation** (**S2_SFI**): triggers the irrigation of a predefined amount of water when the soil water deficit is above a certain threshold. The full irrigation assumes an unlimited amount of water availability. This strategy is expected to consume the maximum amount of water while achieving the yield potential.
3. **Simple Deficit irrigation** (**S3_DI**): triggers irrigation of a predefined amount of water when the soil water deficit is above a threshold which already causes drought stress for the crop. This irrigation strategy is a simple implementation of deficit irrigation. It is expected that **S3_DI** consumes less water than **S2_SFI**, but full irrigation cannot be applied when water availability is constrained or limited. **S3_DI** serves as a non-optimized deficit irrigation strategy which is compared with other optimized deficit irrigation strategies.
4. **Constant supplemental irrigation in a fixed schedule** (**S4_CFS**): realizes a fixed application depth of water for a fixed irrigation interval of days (e.g., 7 days between applications). This deficit irrigation strategy can deal with limited given water volumes but implements a non-optimized strategy which is expected to achieve a low yield.
5. **Optimized deficit irrigation with decision table** (**S5_ODT**): is a closed-loop irrigation control based on information about the available water and the water deficit in the soil. For daily decisions, a decision table is optimized for maximizing water productivity. The optimizer was implemented using Evolution Strategy with Covariance Matrix Adaptation (CMA-ES) for nonlinear function minimization, Version 3.61. Beta [53].
6. **Optimized deficit irrigation with a decision table and phenological stages** (**S6_ODTph**): implements a modified decision table based on the crop response to water stress at the specific phenological stages throughout the growing season. The optimizing process was also implemented using CMA-ES.
7. **Optimized deficit irrigation with Global Evolutionary Technique for Optimal Irrigation Scheduling (GET-OPTIS)** (**S7_GO**): is an open-loop irrigation control that implements a general irrigation calendar which is valid for all growing seasons of a considered time series.

The implementation is based on the tailor-made evolutionary GET-OPTIS algorithm developed by Schütze et al. [38]. This strategy allows for a simpler application in practice than **S5_ODT** and **S6_ODTph** since no information about the water deficit in the soil is required.

**S1_RF**, **S2_SFI**, and **S3_DI** were evaluated using the workflow outlined in Figure 2. The remaining strategies were implemented based on the workflow shown in Figure 3. Consequently, for the optimized strategies the computational demand is significantly higher. Furthermore, **S2_SFI**, **S3_DI**, **S5_ODT**, and **S6_ODTph** strategy require sensor information about either climate and/or soil variables. On the contrary, **S1_RF**, **S4_CFS**, and **S7_GO** strategy are the cheapest and easiest to use.

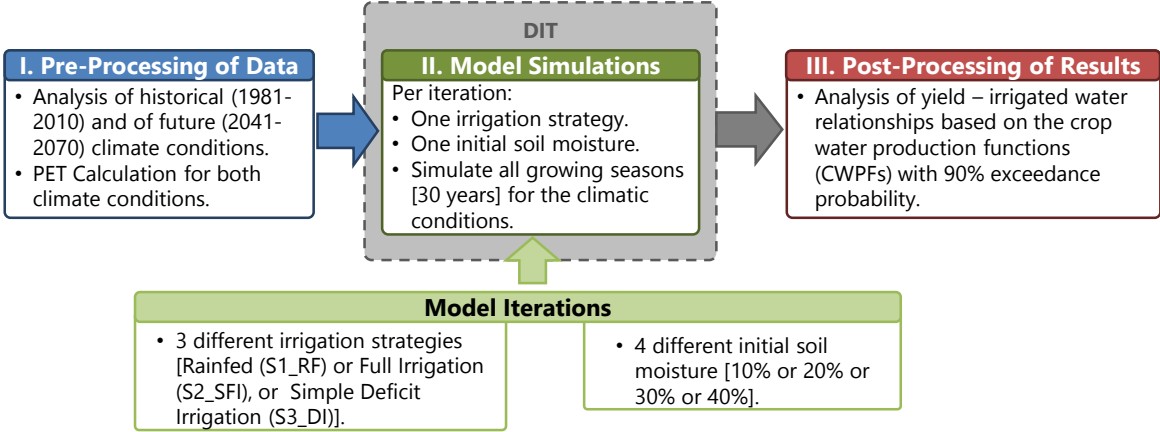

**Figure 2.** Model framework for the basic irrigation strategies (S1_RF, S2_SFI, S3_DI).

### 2.4. Model Framework

The assessment of diverse irrigation strategies was modeled using the DIT [54], an open-source software to analyze the crop yield response to climate and soil variability, as well as water management practices. The DIT considers several irrigation scheduling strategies and different crop models such as AquaCrop [18] and Soil-Water Balance (SWB) [55]. The stochastic relationship between simulated yield and irrigated water also known as Stochastic Crop Water Production Functions (SCWPF), the main result of the DIT, is an effective tool for risk analysis on irrigation demand [37]. The framework used in the DIT was applied and validated in different field studies (e.g., Grundmann et al. [56], Schütze et al. [57], and Gadédjisso-Tossou [54]).

For this study, the Soil-Water Balance Model (SWB) [55] was combined with seven different irrigation strategies available in the DIT. The SWB model is a relatively simple model that simulates the yield response based on the water deficits in the soil storage. The choice of this model was to avoid confounding in the interpretation of the results with other complex models and can be undertaken in a future study with more available data. Despite its simplicity, the model demonstrated reliable performance in previous studies (e.g., Rao et al. (1988 [58], 1992 [59]), Panigrahi et al. [60], Khan et al. [61] and Gassmann et al. [62]). The ability of the model to be really responsive to hydroclimatic variability in one of the inherent strengths and needs in choosing this modeling system.

Each irrigation strategy implementation followed a certain framework. For the first three irrigation strategies (S1_RF, S2_SFI, S3_DI), the workflow considered in this study is shown in Figure 2. The first framework mainly consists of three phases: I. Pre-Processing of the data for both the historical and future climatic conditions, including the calculation of daily potential evapotranspiration; II. Model simulations for multiple configurations within the DIT for the growing seasons within the climatic conditions assuming a specified initial soil moisture condition; III. Post-processing of the results by analyzing the SCWPFs within the 90% exceedance probability. For the other strategies (S4_CFS, S5_ODT, S6_ODTph, S7_GO) the framework outlined in Figure 3 was used.

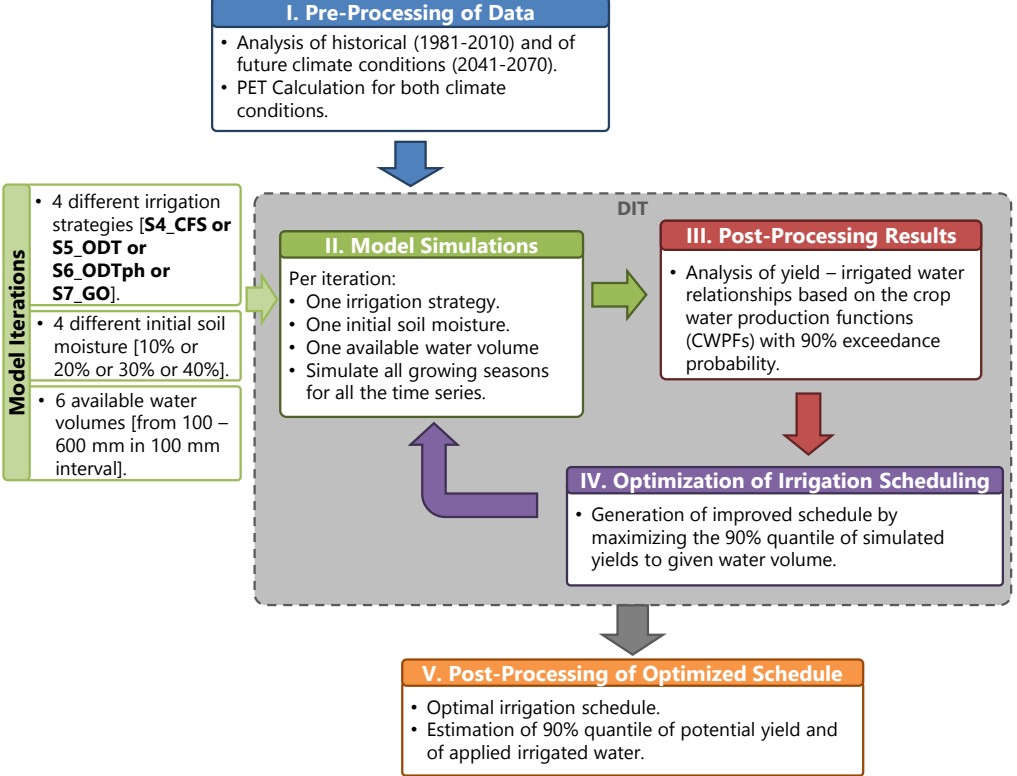

**Figure 3.** Model framework for the optimized irrigation strategies (S4_CFS, S5_ODT, S6_ODTph, S7_GO).

Similar to the first three phases shown in Figure 2; an additional iteration for limited available water volume between 100 to 600 mm. The incorporation of optimization phases IV and V to account for robustness as the optimizer maximizes a larger quantile (e.g., 90%) of the yields of the simulated scenarios to ensure high water productivity.

The optimization step is implemented using various global, computational demanding optimization techniques (Section 2.3). This proposed framework allows for the risk analysis and assessment of both historical and climate change scenarios within different conditions of water availability. The product of these model simulations is the SCWPFs, (i.e., the stochastic relationship of simulated yield and irrigated water), which represent the risk pattern for a specific irrigation location and certain initial and boundary conditions [63].

*2.5. Experimental Design of Model Simulations*

The present study undertook multiple simulations for each site and climatic conditions. The experimental design of each simulation follows the sequence shown in Figure 4. This sequence comprised of four steps: (1) The model simulations were carried out resulting in 108 simulation results per site for each climatic conditions, (2) The analysis of these results based on the location within the US Corn Belt, (3) The analysis of the main hydroclimatic parameters to better understand the changes on the simulated yields, (4) The final evaluation of each strategy for both historical and future climatic conditions as well as performance metrics of the irrigation strategy model based on the reported annual yields in the historical climatic conditions.

As was described previously, the model simulations for each study site comprised of seven different irrigation strategies, each analyzed with four different initial soil moisture conditions. For the first three irrigation strategies (S1_RF, S2_SFI, S3_DI) each strategy-soil moisture iteration was modeled with only one available water volume and for the remaining four strategies (S4_CFS, S5_ODT, S6_ODTph, S7_GO), each strategy-soil moisture iteration was modeled with six different available

water volume to irrigate. This resulted in 108 simulations for each site or a total 1836 simulations for each 30 year long climatic conditions in the US Corn Belt. These results were then grouped depending on the location of the site within the US Corn Belt, as each region had different implementation of irrigation. The Western Corn Belt reported the most irrigation applied primarily in Kansas and Nebraska. The Eastern Corn Belt reported very little irrigation. In the next step, a hydroclimatic analysis of the main parameters (i.e., temperature, precipitation, and potential evapotranspiration) was carried out. This aimed to further understand the changes between both historical and future climatic conditions and the impacts on yield and water resources availability. The last step was the evaluation of the model simulations, first for the performance of the model based on the annual yields on the historical climatic conditions and lastly, a comparison of the best performing strategy (i.e., higher potential yield with less applied water) within all the strategies considered.

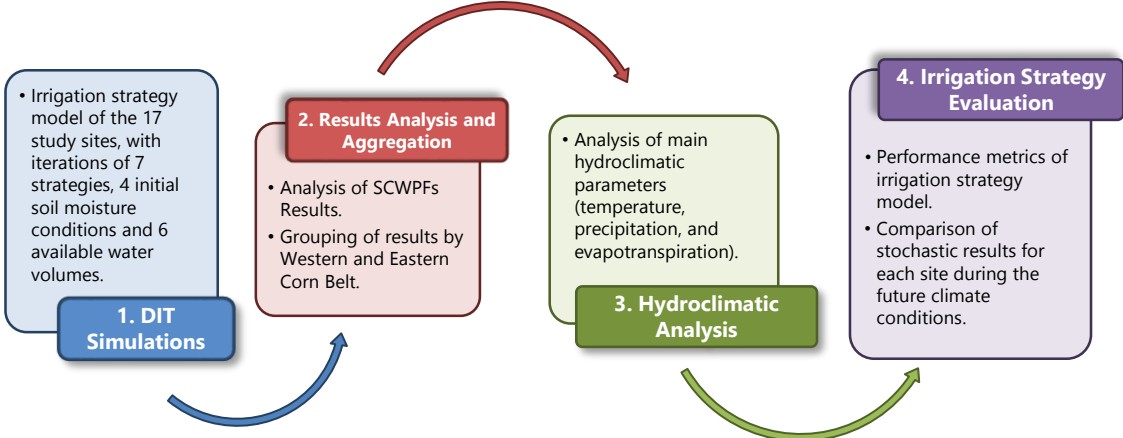

**Figure 4.** Main Steps of the Experimental Design of Model Simulations.

### 2.5.1. Hydroclimatic Variability Analysis

The hydroclimatic data analyzed for each site was daily precipitation, maximum and minimum temperature, solar radiation and the estimated potential evapotranspiration. The Corn Belt, particularly the Eastern region, is known for its suitability for rainfed agriculture, nevertheless previous studies (e.g., Alter et al. [64], Karl et al. [10], Gunn et al. [25], Pryor et al. [65], Djaman et al. [23] and Dai et al. [66]) analyzed the impact of the past and projected hydroclimatic changes on the food production in the Midwest US. To help offset the reliance on rainfed agriculture, studies such as Van Dop et al. [67] project an increase in the number of counties within the U.S. where the optimal yield could be improved by the application of irrigation, making this technology a profitable investment. For both historical and future climatic conditions, the main hydroclimatic parameters (average temperature, total precipitation and potential evapotranspiration) were analyzed within the months of April to September which comprised the common 150 days of the growing season within the US Corn Belt. The differences between each region of the Corn Belt and each time series were analyzed.

### 2.5.2. Model Performance Metrics

Model evaluation metrics assess goodness of fit between model predictions and data. One widely used performance metric is the Mean Absolute Error (MAE). The MAE compares simulated yield with the relative observed yields for each site. This was calculated as follows:

$$MAE = \frac{1}{n}\sum_{i=1}^{n}|Y_s - Y_o| \tag{1}$$

where $Y_s$ is simulated yield and $Y_o$ is the reported data. The advantage of using MAE is not only that it is easy to interpret but also allows a comparison with previous studies (e.g., Liu et al. [43] and Niyogi et al. [1]) where the same data sets were evaluated with different crop models.

### 2.5.3. Evaluation of Irrigation Strategies

The evaluation of the model simulations of each irrigation strategy was based on two limits to assure a true optimization of irrigation application. These two limits were: (i) The optimal conditions for irrigation application based on the simple deficit irrigation estimation, which defines the maximum volume of irrigated water that is not exceeded and (ii) the optimal rainfed conditions, which defines the minimum optimal yield that needs to be achieved. As a result, the simulated results must display a higher potential yield than the rainfed (S1_RF) and higher savings (less irrigated water applied) as compared to the simple deficit irrigation (S3_DI) strategy.

## 3. Results and Discussion

### 3.1. Hydroclimatic Variability Analysis

The monthly distribution within the growing season of hydroclimatic parameters: temperature, precipitation, and evapotranspiration in the historical and future climatic conditions are shown in Figure 5, which consists of two sets of plots. The left side (plots a, b, and c) show the historical climatic conditions and the right side (plots d, e, and f) show the future climatic conditions. Each plot is described by a colored central box (blue for Eastern, and red for Western Corn Belt) that represents the distribution of the data where the first and third quartile are the lower and upper boundary lines respectively and the central point indicates the median. The vertical lines extending from the box indicate the data outside of the main quartiles. The outsiders represent the variability within the years and the dotted lines represent the average trend for each parameter in the sites located in each region of the US Corn Belt.

Considering the changes in temperature in both historical and future climatic conditions, the variability within the sites in the Western Corn Belt is higher than in the sites located in the Eastern Corn Belt. The trend in the future growing seasons seems to change, where it is expected a warmer and earlier spring and lower temperatures during summer. The warming in the early months has already changed the dates of the growing season in each county independently by around 12 days longer than it was a century ago [68]. It is estimated an overall warmer temperature during the growing season which could affect not only corn agriculture but other productive crops. A more intensive analysis of the temperature in the Midwest US performed by Dai et al. [66] showed that the early growing season average temperature increased at a rate of 0.15 °C/decade overall, showing different trends for minimum and maximum temperature as well as maximum solar radiation.

Precipitation in both historical and future climatic conditions showed to have a wider inter-annual stochastic variation (i.e., the data outside the central box show a significant increase) from site to site from all the other climate parameters. This randomness could be explained by the increase in short duration heavy rainfalls that are predicted across the US Corn Belt. These extreme rainfall events show an increasing trend, even though the average precipitation showed a general decreasing trend from April to July in the Eastern Corn Belt and throughout all the growing season in the Western Corn Belt. Van Wart et al. [69] demonstrated that the sites located in the Western Corn Belt were more frequently subjected to an episode of transient and erratic rainfall in the critical development stage leading to extra fieldwork, such as drying crops or even bigger yield lost.

The future climatic conditions show lower values of average solar radiation (from the NARCCAP data set) which results in the lower estimation of potential evapotranspiration based on the Penman-Monteith equation. The maximum obtainable yield is reached only when enough water is provided to satisfy crop requirement; hence, irrigation is triggered when the crop has not enough water to meet the maximum evapotranspiration requirements [19]. Climatic variability between

different locations have a significant impact on the yield production, due to the interaction of precipitation, potential evapotranspiration and plant growth requirements. In particular, the amount of water required by maize throughout the growing season depends on the evaporative demand of the atmosphere and water availability [18]. The difference between the two climatic conditions (historical vs future) coincides with the temperature and precipitation changes which connote significant variations of solar radiation, wind velocity, and humidity. Further studies such as that by Basso et al. [70] analyzed the impact of this change in evapotranspiration for the current seeds used in the Corn Belt and concluded that the current high yield can be obtained when the water supply is constantly between 500 to 700 mm for the growing season.

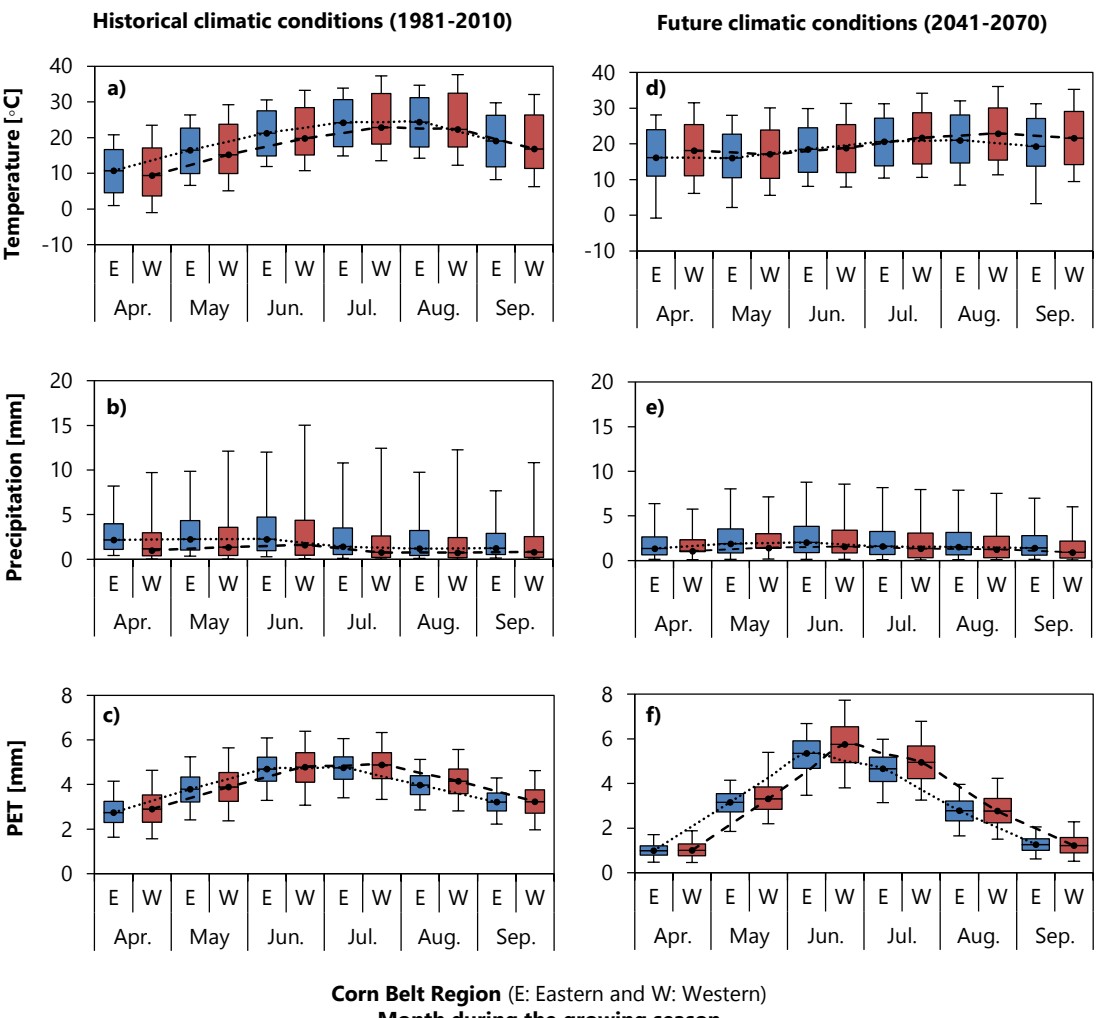

**Figure 5.** Box-and-whisker plots of the monthly trends of the hydroclimatic parameters in the historical (**left side a to c**) and future (**right side d to f**) climatic conditions for the sites located in the Eastern Corn Belt (blue) and the sites located in the Western Corn Belt (red).

### 3.2. Model Performance Metrics

The mean absolute error (MAE) was used to assess the performance of the irrigation strategy model.

The MAE (Table 2) summarized the overall performance of the model for each site. For the sites located in the Eastern Corn Belt, the model performed slightly better with an average MAE of 1.7 tons/ha where for the sites located in the Western Corn Belt the average MAE is 2.02 tons/ha. Previous studies by Liu et al. [43] and Niyogi et al. [1] used the same onsite climatological data with

three different crop models: the Hybrid-Maize [33], the Decision Support System for Agrotechnology Transfer (DSSAT) [33] and the Integrated Science Assessment Model (ISAM) [63] in order to assess the impact of model complexity on simulated corn yield in response to climate change. The accuracy of the implemented model in this study shows similar prediction accuracy to the Hybrid-Maize model which was the best of the three models used and was also the simplest crop model. These 17 case studies results provide additional confidence in using the Deficit Irrigation Toolbox to achieve useful model responsiveness to high hydroclimatic and spatial variability.

**Table 2.** Mean absolute error (MAE, tons/ha) of simulated corn yields in the historical climatic conditions.

| Site | Code | Observed Yield [tons/ha] | | Predicted Yield [tons/ha] | | Mean Absolute Error [tons/ha] |
| | | Average | Std. Dev | Average | Std. Dev | |
| --- | --- | --- | --- | --- | --- | --- |
| New Madrid | W3 | 9.72 | 1.23 | 8.87 | 1.64 | 1.78 |
| Topeka | W2 | 7.77 | 1.46 | 7.53 | 1.83 | 0.94 |
| Kirksville | W1 | 6.96 | 2.10 | 7.24 | 1.52 | 2.11 |
| Columbus | W8 | 9.08 | 1.85 | 9.30 | 2.04 | 2.16 |
| Brookings | W5 | 7.25 | 2.07 | 7.33 | 1.87 | 2.47 |
| Grand Forks | W7 | 5.96 | 1.67 | 6.86 | 1.87 | 1.94 |
| Iowa City | W6 | 8.89 | 2.16 | 7.52 | 1.77 | 2.23 |
| Olivia | W4 | 9.51 | 2.08 | 9.72 | 1.83 | 1.49 |
| Rochester | W9 | 9.51 | 2.07 | 8.99 | 2.39 | 2.11 |
| Baraboo | E4 | 8.42 | 1.37 | 8.50 | 1.48 | 1.40 |
| Beloit | E6 | 8.87 | 1.52 | 8.91 | 1.84 | 1.21 |
| DeKalb | E5 | 9.96 | 1.62 | 9.84 | 2.59 | 1.98 |
| Rensselaer | E7 | 8.93 | 2.00 | 8.61 | 2.20 | 1.98 |
| Tuscola | E8 | 9.69 | 1.78 | 9.29 | 2.27 | 1.76 |
| Huntington | E3 | 8.83 | 1.70 | 8.29 | 2.14 | 1.52 |
| Marysville | E1 | 8.49 | 2.01 | 9.14 | 1.81 | 2.04 |
| Toledo | E2 | 9.39 | 1.68 | 9.55 | 2.24 | 1.69 |

### 3.3. Results of Evaluation of Irrigation Strategies

Yield development is impacted by water stress, which was different across sites and the historical and future climatic conditions. Following the experimental design, for every study site, the stochastic crop water production functions (SCWPFs) were estimated based on the limited available water volumes. Figure 6 shows an example of the simulation results for the site in Topeka, KS (W2) for both time series with initial soil moisture of 20%. The different shades of grey in Figure 6 represent the level of optimization achieved by the strategies, where the SCWPF found in the white area are the optimal simulations based on the evaluation metrics (Section 2.5.3). The results in the grey areas show only water saving compared to the full supplemental irrigation.

The results in all 17 study sites indicate that water availability was enough in both historical and future climatic conditions to grow corn under rainfed conditions with a very low yield and with high variability between years. Also, all strategies show significantly different SCWPF in the different soil moisture analyzed. The impact of the hydroclimatic variability between the historical and future climatic conditions is shown simply by the simulated yield with the rainfed strategy (S1_RF). Where the potential yield decreased around 20%, highlighting the need for future optimized irrigation strategies that consider limited available water.

To summarize the main findings of the evaluation of the diverse irrigation strategies, for the case of the constant supplemental irrigation in a fixed schedule strategy (S4_CFS) an improvement in yield can be seen only above rainfed conditions, although small water savings compared to deficit irrigation strategies can be seen only in wet soil conditions (i.e., above 30% initial soil moisture). The decision tables and GET-OPTIS optimizers (S5_ODT, S6_ODTph, S7_GO) showed better results within all the study sites. Both optimizers increased water productivity when compared to non-optimized irrigation strategies. GET-OPTIS (S7_GO) showed better results for wet soil conditions with higher precipitation variability and the Decision Tables performed better for dry soil conditions with high precipitation variability. In particular, the optimized deficit irrigation with decision table strategy (S5_ODT) and

with phenological stages strategy (S6_ODTph) showed improvement in all soil conditions in the historical climatic conditions with more than 50 mm savings of irrigated water. The results during the future climatic conditions demonstrate that the variability within the hydroclimatic parameters affects differently each location, resulting in variable water demands for the entire region.

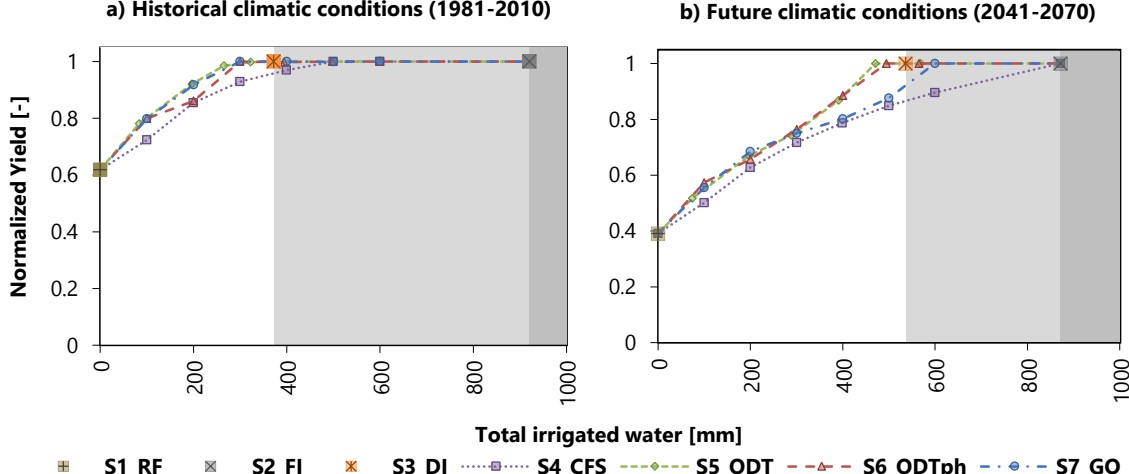

**Figure 6.** 90% Quantile of Stochastic Crop Water Production Functions for the site in Topeka, KS (W2) for (**a**) historical and (**b**) future climatic conditions with initial soil moisture of 20%.

The 90% quantile of Stochastic Crop Water Production Functions (SCWPFs) of the study sites are significantly different at several levels of irrigation with a proportional increasing trend with the available volume, where a higher level of irrigated volume is required for dry soil conditions. In wet soil conditions (40% initial soil moisture), the irrigation strategies have no impact on the yield, showing the importance of plant-soil moisture data to achieve precise irrigation scheduling. In all locations, stochastic variability between years showed to be higher at low levels of irrigation. This was evidently improved by GET-OPTIS and Decision Tables optimizers. These optimizers performed better for the climates with low and variable rainfall were the irrigation can become the stable source of water for the plant at the most important stages of crop development, the vegetative and reproductive. Plots of each soil moisture-initial conditions for all the 17 study sites are presented in Supplementary Material SII.

Figure 7 shows the evaluation of all the simulation results during future climate conditions. The optimal performing strategy, based on the highest achievable yield with the lowest applied water, is shown for each soil moisture conditions considered.

A detailed evaluation of the optimal irrigation strategy is shown in Supplementary Material SIII, where the percentage of improved yields and water savings are shown based on the two limits set to ensure the optimization of irrigation strategies. Regarding the improvement in the potential yields, Tuscola, IL (E8) for up to 85% in dry soil conditions and 21% for wet soil conditions. Topeka, KS (W2) also show the largest improvement, with more than 70% in dry soil conditions and 12% in wet soil conditions. Regarding irrigated water savings, Huntington, IN (E3) had more than 90 mm in savings in dry soil conditions and Rensselaer, IN (E7) and Toledo, OH (E2) had around 100 mm in savings for wet soil conditions. Grand Forks, ND (W7) and Columbus, NE (W8) showed improvements above the 100 mm for dry soil conditions and wet soil conditions respectively. In both locations (W7 and W8) irrigation technologies are already being implemented, highlighting the potential of deficit irrigation technologies to maximize water productivity. The sites located in the center of the Corn Belt (i.e., Rochester, MN (W9) Baraboo, WI (E4), and Beloit, WI (E6)) in wet soil conditions showed no improvement in yields compare to the rainfed strategy S1_RF.

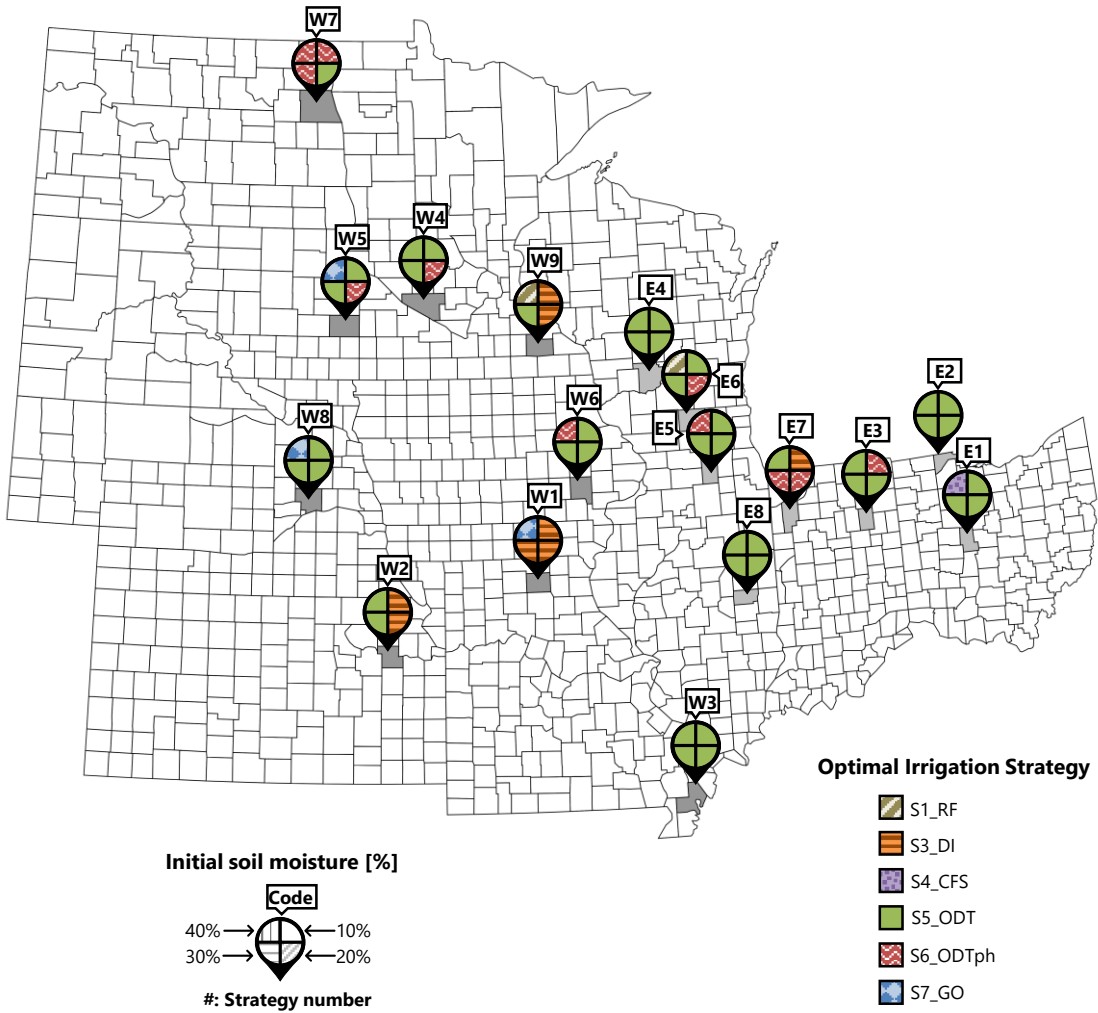

**Figure 7.** Optimal irrigation strategy for every initial soil moisture [10–40%] for future climatic conditions.

### 3.4. Summary of Discussion

Recommendations for full or limited irrigation differ in practice and literature, hence the evaluation of a wide range of irrigation strategies was carried out. Results highlight the potential of deficit irrigation to be beneficial for crop yield, yet also considering rainfed and supplemental irrigation approaches. The optimization of deficit irrigation strategies increased water productivity for the historical climatic conditions as well and showed potential to conserve water and improve yield productivity for the future climatic conditions. In years of predicted water scarcity, yields of at least 60% could be achieved with 200 mm of irrigation water at very high reliability when optimization strategies are used. In the same conditions, the rainfed strategy achieved less than 40% in the study sites of the Corn Belt. The simulated level of irrigated water coincides with the optimum crop water productivity values reported for irrigated maize by Zwart et al. [29], demonstrating the high risk of non-optimized schedule for sites with high climate variability. Based on the performance of the strategies, the future investment in irrigation equipment can be expected to happen primarily in the sites located in the Eastern Corn Belt within Ohio and Great Lakes water resource region. The sites located in the Western Corn Belt, where there is already irrigation, are expected to continue adapting for efficient irrigation practices to minimize the impact of the water demand on other vital demands. The exception of Iowa and Minnesota in the Upper Mississippi water resource region where the implementation of irrigation did not show a significant improvement of the rainfed agriculture for it to be considered a profitable investment base merely on the gains in simulated yields.

*3.5. Limitations*

The irrigation strategy model has several limitations. The first concern was that the observed data available was for only rainfed conditions, therefore the model showed slightly different results for the counties where irrigation is already being applied. Additionally, the simplicity of the SWB has inherent limitations predicting crop yields for the different growing season dates. Another simplification of our experimental designed is the choice of 10% ranges of initial soil moisture percentage due to the lack of data in different temporal scales. Despite such limitations, the model framework, as well as the experimental analysis proved its high usefulness and big potential for further specialized studies.

## 4. Conclusions

This study aimed to analyze the hydroclimatic variability at different temporal scales and to evaluate supplemental and deficit irrigation optimizers under potential water scarcity conditions over US Corn Belt, where rainfed conditions are expected to change and adaptations strategies are needed. The experimental design using the DIT were used to integrate different irrigation strategies into a parsimonious crop model that boosts crop efficiency and reduce the impact on water resources in a changing climate. The previously favorable hydroclimatic conditions in the Corn Belt for rainfed agriculture are estimated to change, opening the opportunity for mitigation strategies. The results show a decreasing trend in seasonal precipitation but an increasing trend in temperature and potential evapotranspiration for future growing seasons. The spatial and temporal variability of the precipitation changes shown by the increased stochastic variations suggests the need for additional catchment capacities and an increase in the water demand for agricultural production affecting all the other water demands. Higher hydroclimatic variability implies higher risks yield reduction, consequently, the simulations showed the great potential of deficit irrigation optimization strategies to increase the water and yield productivity for future growing seasons. The Decision Tables and GET-OPTIS optimizers showed good results for the study sites, GET-OPTIS showed better results for wet soil conditions with higher precipitation variability and the Decision Tables performed better for dry soil conditions seasons with high precipitation variability. The regionalization of more studies of areas surrounding the studied sites where a more complex crop model with specialized soil and climate data can be done based on the optimal irrigation scheduling strategy produced by this study.

**Supplementary Materials:** The following are available at http://www.mdpi.com/2073-4441/11/12/2447/s1, The supplementary material consists of three parts: Supplementary Material SI: Detailed information about the weather station and important seeding and harvest date for each study site during each climatic conditions; Supplementary Material SII: The extensive simulation results for all study sites in the different initial soil moisture conditions; Supplementary Material SIII: The detailed evaluation of optimal irrigation strategy for future climatic conditions.

**Author Contributions:** This research was done in collaboration of all authors. M.E.O.A. worked on the data preparation, development and application of the methodology, analysis, and validation of results, and writing the manuscript. N.S. worked on the conceptualization, development of methodology, supervision, and manuscript review. D.N. worked on data integration, supervision, and manuscript review.

**Funding:** This research was funded by the Technische Universität Dresden, by means of the Excellence Initiative by the German Federal and State Governments.

**Acknowledgments:** This research was carried out within the International Research Training Group "Resilient Complex Water Networks". The authors would like to express our gratitude for all the technical support from Technische Universität Dresden and Purdue University. We wish to thank the North American Regional Climate Change Assessment Program (NARCCAP) for providing the data used in this paper. NARCCAP is funded by the National Science Foundation (NSF), the U.S. Department of Energy (DoE), the National Oceanic and Atmospheric Administration (NOAA), and the U.S. Environmental Protection Agency Office of Research and Development (EPA). To the anonymous reviewers for their constructive criticisms and helpful suggestions.

**Conflicts of Interest:** The authors declare no conflict of interest.

## Abbreviations

The following abbreviations are used more than once in this manuscript:

| | |
|---|---|
| CWP | Crop Water Productivity |
| US | United States of America |
| SCWPF | Stochastic Crop Water Production Functions |
| SWB | Simple Soil-Water Balance Model for Irrigated Areas |
| CMA-ES | Evolution Strategy with Covariance Matrix Adaptation |
| GET-OPTIS | Global evolutionary Technique for Optimal Irrigation Scheduling |
| S1_RF | Rainfed irrigation |
| S3_DI | Simple deficit irrigation |
| S4_CFS | Constant supplemental irrigation in a fixed schedule |
| S5_ODT | Optimized deficit irrigation with decision table |
| S6_ODTph | Optimized deficit irrigation with decision table with phenological stages |
| S7_GO | Optimized deficit irrigation with GET-OPTIS |
| MAE | Mean Absolute Error |

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
