# Peer review of "Evaluation of Hydroclimatic Variability and Prospective Irrigation Strategies in the U.S. Corn Belt"

_water, doi:10.3390/w11122447_

Round 1

Reviewer 1 Report

   Please see the attachment file

Reviewer 2 Report

Overall this is a relatively strong paper that with some small clarifications should be suitable for publication. 

Irrigation scheduling is not defined until line 84. It should be defined earlier.

Line 94 introduced this actual study this should appear ahead of the broad literature review on irrigation scheduling. 

110-113 There has been irrigation expansion in the corn belt outside of this region, but generally in areas with soils with low water capacity. 

Section 2.2 Was soils data used in this study? The water holding capacity of the soil would be critical for several of the model runs that the paper proposes, since this would require using stressed conditions calculations from the FAO-56 method (see chapter 8). In section 2.4 it seems like soils data is incorporated into the DIT model, but no source of soils data is listed. This requires clarification.

321 Easter corn belt. Only young children believe in the Easter corn belt!

For measures of water productivity please clarify whether total CWP is being calculated (i.e. of yield:ET) or only productivity of irrigation water (ie yield:irrigation water, or blue water CWP)

Round 2

Reviewer 1 Report

The authors have improved the paper according to the reviewer comments, so it can be accepted for publication